

# Using historical dip net data to infer absence of flatwoods salamanders in stochastic environments

George C. Brooks and Carola A. Haas

Department of Fish and Wildlife Conservation, Virginia Polytechnic Institute and State
University (Virginia Tech), Blacksburg, Virginia, United States

## ABSTRACT

Local extinction and undetected presence are two very different biological
phenomena, but they can be challenging to differentiate. Stochastic environments
hamper the development of standardized monitoring schemes for wildlife, and make
it more challenging to plan and evaluate the success of conservation efforts.
To avoid reintroductions of species at risk that could jeopardize extant populations,
managers attempting translocation events require a higher level of confidence that a
failure to confirm presence represents a true absence. For many pond breeding
amphibians, monitoring of the breeding population occurs indirectly through
larval surveys. Larval development and successful recruitment only occurs after a
sequence of appropriate environmental conditions, thus it is possible for a breeding
population of adults to exist at a site but for detectability of the species to be
functionally zero. We investigate how annual variability in detection influences
long-term monitoring efforts of Reticulated Flatwoods Salamanders (*Ambystoma
bishopi*) breeding in 29 wetlands in Florida. Using 8 years of historic dip net data, we
simulate plausible monitoring scenarios that incorporate environmental stochasticity
into estimates of detection probability. We found that annual variation in
environmental conditions precluded a high degree of certainty in predicting site
status for low-intensity monitoring schemes. Uncertainty was partly alleviated by
increasing survey effort, but even at the highest level of sampling intensity assessed,
multiple years of monitoring are required to confidently determine presence/absence
at a site. Combined with assessments of habitat quality and landscape connectivity,
our results can be used to identify sites suitable for reintroduction efforts.
Our methodologies can be generally applied to increase the effectiveness of surveys
for diverse organisms for which annual variability in detectability is known.

## INTRODUCTION

Local extinction and undetected presence are two very different phenomena, but they can
be challenging to differentiate. There has been much attention devoted to detection
probability, and especially to the challenges raised when working with rare and cryptic
species in stochastic environments (*Lebreton et al., 1992*; *MacKenzie et al., 2002*;
*Royle, Nichols & Kéry, 2005*; *Pacifici, Dorazio & Conroy, 2012*; *Kellner & Swihart, 2014*;

Corresponding author
George C. Brooks, boa10gb@vt.edu

*Moore et al., 2014*; *Specht et al., 2017*; *Folt et al., 2019*). The detectability of many species varies with environmental conditions; annual variation in weather can strongly affect the optimal time of year to survey, and the overall likelihood of detecting the target species (*Field, Tyre & Possingham, 2005*; *Jackson et al., 2006*; *McConville et al., 2009*; *Rizzo et al., 2017*; *Shaffer, Roloff & Campa, 2019*). If the factors influencing detectability are not known, one can never be confident that failure to detect constitutes a true absence, and the development of standardized monitoring schemes will be severely hampered (*Penteriani et al., 2005*; *Bried & Pellet, 2012*; *Gervasi et al., 2014*; *Bellier, Kéry & Schaub, 2016*; *Crone, 2016*).

Species with complex life cycles pose an additional challenge because different life-stages may require different sampling methodologies, or may only be available for capture at particular times of the year. Many pond-breeding amphibians are fossorial for most of the year, migrating to and from breeding sites annually in response to seasonal cues. For such species, it is nearly impossible to avoid monitoring techniques that rely on specific environmental conditions to be effective. Factors known to directly influence detectability are often unknown, difficult to assess, or poorly understood. Thus in many instances, monitoring schemes cannot be strategically tailored to maximize success with any degree of precision.

Amphibian monitoring typically occurs at breeding wetlands, either through drift-fences that intercept arriving individuals, dip-net surveys for aquatic larvae, or egg-mass counts (*Heyer et al., 1994*; *Wells, 2007*; *Wilkinson, 2015*). Owing to the high labor cost of continually running a drift-fence however, monitoring of adult populations is often sporadic or entirely absent. Population assessments based solely on dip-net data are common, yet fraught with uncertainty (*Penteriani et al., 2005*; *Bried & Pellet, 2012*; *Gervasi et al., 2014*; *Bellier, Kéry & Schaub, 2016*; *Crone, 2016*). Because larval development occurs only in years with suitable weather and hydrological conditions, it is possible for a breeding population of adults to occur but for the detection probability of larvae to be zero. Failure to detect may simply represent a bust recruitment year, offering no information as to the health of the population. In some cases it should be easy enough to realize that the species would be undetectable in a given year (extreme drought), but certain patterns of pond filling and drying and refilling, or flooding carrying in fish predators, may also cause complete egg or larval mortality without being obvious (*Semlitsch, 1987*; *Dodd, 1993*; *Taylor, Scott & Gibbons, 2006*).

In some cases, the repercussions of failing to detect an organism are considerably more severe than incorrectly documenting a presence, *i.e.*, false negatives are more problematic than false positives (*Hauser & McCarthy, 2009*; *Charney, Kubel & Eiseman, 2015*; *Veale & Russello, 2016*). Conservation reintroductions are a useful example of the risk imbalance between type-1 and type-2 errors. Failure to detect a target organism may result in reintroductions that could jeopardize extant populations (*Kéry & Schmidt, 2008*). Introducing animals to an extant population can have positive (*Hedrick & Fredrickson, 2010*) or negative (*Laikre et al., 2010*) impacts on genetic viability, and as such a clear distinction must be made between establishing new populations and augmenting existing ones. Identifying suitable, unoccupied sites for establishing new populations can

be hampered by environmental stochasticity. Similarly, evaluating the relative success of reintroduction efforts is made challenging by low or uncertain detectability brought on by environmental variation. Thus there is a clear need for approaches that can confidently predict site status and determine the fate of individuals following reintroduction under a range of environmental conditions.

We have monitored hydrology and worked with partners to restore suitable vegetation in wetlands across Eglin Air Force Base, in hopes of restoring Reticulated Flatwoods Salamanders (*Ambystoma bishopi*) to a greater proportion of their former range. Procedures for allowing an existing population to expand into improved habitat, augmenting an existing population, or conducting a re-introduction are very different (and for a protected species would require different levels of approval from regulatory agencies). Assessing whether a translocation action would result in augmentation *vs.* re-introduction depends on knowing whether there is an extant population or not. Here we investigate how annual variability in detection influences long-term monitoring efforts. Using historic dipnet data we simulate a range of scenarios that incorporate environmental stochasticity into estimates of detection probability and evaluate the efficacy of different monitoring schemes. In the face of environmental uncertainty, identifying potential reintroduction sites requires high survey effort over a period of years to confidently declare absence. Our procedure can be adapted for any species when there are data available for detection probability over a series of years that encompass a meaningful range of environmental conditions.

## MATERIALS & METHODS

### Study site and species

Our study was conducted on Eglin Air Force Base (Eglin), Okaloosa and Santa Rosa counties, Florida, USA from 2009 to 2017. All field work was authorized by the Department of Defense and the Jackson Guard Natural Resources Office. Eglin is a large military installation (188,459 ha) primarily consisting of actively managed longleaf pine-dominated sandhills (approximately 145,000 ha) interspersed with treeless open test ranges, pine plantations, and mesic flatwoods. The study focused on 29 ephemeral wetlands that have been monitored intensively for Reticulated Flatwoods Salamander larvae since 2009. Reticulated Flatwoods Salamanders are mole salamanders in the family Ambystomatidae and are federally listed as endangered, with restoration efforts underway and reintroduction efforts currently being planned. Individuals migrate in the fall to dry basins, and deposit eggs terrestrially (*Palis, 1997*). Embryos develop so that if winter rains inundate the ponds after several weeks or months, the eggs hatch into aquatic larvae. Wetlands were surveyed for salamander larvae using standard dipnet sampling techniques (*Heyer et al. 1994*; *Bishop et al., 2006*; *Wilkinson 2015*) between 2009 and 2018. Surveyed wetlands range in size from 0.1 ha to 20.9 ha and consist of shallow depressions which fill in the autumn and typically remain inundated throughout the spring (*Chandler et al., 2016*). To account for imperfect detection, at least three surveys were conducted annually, spaced one per calendar month. Surveys were conducted during the

months of January until May, and survey effort was standardized to 30 min for each occasion (see *Brooks et al., 2019* for full description of field methods).

## Statistical approach

Using posterior distributions of year-specific detection probabilities derived from a recent occupancy study (*Brooks et al., 2019*), we investigated the impact of environmental stochasticity on long-term monitoring of Reticulated Flatwoods Salamanders. Over the 8-year study period, detection probabilities conditional on a site being occupied ranged from 0.02 to 0.88 for a single survey. We adopted a bootstrapping approach, whereby years were randomly drawn from the dataset with replacement to construct a decade-long series of potential detection probabilities. We iterated this procedure 1,000 times to generate a sample of hypothetical time-series, reflecting our uncertainty in detection given annual environmental fluctuations. A decade was chosen partly to reflect the study duration from which the original data were drawn, and also to approximate the longevity of the species (*Brooks, 2020*), *i.e.* if our aim is to determine whether Reticulated Flatwoods Salamanders have been extirpated from a site, we would need to continue monitoring until we were confident that no breeding adults remained on the landscape.

We first calculated the detection probabilities and associated uncertainty with increasing survey effort (up to 10 surveys) conducted within a single year. We combined the estimates from all years for which detection was available, corresponding to a situation in which the relationship between environmental conditions and detectability are unknown. We then estimated the likelihood of detecting larvae given a site is occupied under four levels of survey effort. Scenarios ranged from one to four surveys per year and extrapolated out to 10 years of continuous monitoring for each of the 1,000 hypothetical detection histories. Cumulative detection probabilities for multiple surveys within years were calculated by taking the exponent of the probability of failing to detect an animal, $(1 - p)^n$, where p is the detection probability for a given year and n is the number of surveys. Cumulative detection probabilities across years are simply one minus the product of each year-specific estimate. This approach requires that a site's occupancy status does not change over the course of monitoring, but given the high site fidelity of this species (*Brooks et al., 2019*), we consider there to be a negligible probability of emigration from the focal site over that time (*i.e.* no turnover). From the 1,000 scenarios we derived 90% quantiles for detection probabilities to quantify the uncertainty in site status under different levels of sampling intensity and duration. All analyses were conducted in R version 4.10 (*R Core Team, 2021*). All of the data and code used to perform the analysis and generate the figures are provided as Supplemental Material.

## RESULTS

We found that in the face of environmental uncertainty, increasing the number of intensive, 30-min surveys in a single year could never achieve detection probabilities above 95% with confidence (Fig. 1). Unsurprisingly, given such tremendous annual variability in detection and no knowledge of the environmental conditions in a particular year, an assessment of presence/absence cannot be made without multiple years of surveying.

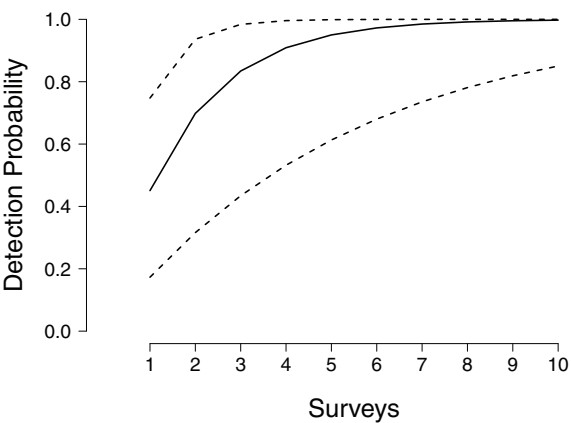

**Figure 1 Annual detection probabilities of larval flatwoods salamanders as a function of survey effort.** The solid line indicates the median detection probability for a single year of monitoring, up to 10 surveys per year. Surveys comprise 30 min of dipnetting effort, typically shared between multiple surveyors, concentrated in areas of herbaceous vegetation. The dashed lines represent bootstrapped quantiles (0.25–0.75) given annual fluctuations in detectability.

Although median detection probability reached 99% after six surveys, the lower bound for detection probability remained below 90% even after 10 surveys (Fig. 1), because during drought years the ability to detect larvae can become next to impossible. In contrast the upper confidence limit for detection approaches unity after only three surveys, reflecting years where conditions for detecting larvae are optimal.

We further found that even when multiple years of surveys are conducted, a high degree of survey effort within years is necessary to achieve high detection rates (Fig. 2). If sites are sampled only once per year, surveys must be carried out for 5 years on average to obtain a detection probability above 95%, and 7 years to obtain a detection probability above 99% (Fig. 2). Again, given uncertainty in the relationship between environmental conditions and detectability, there are wide confidence bands on these estimates, whereby 99% probabilities could be achieved in 3 years or could take a decade, depending on the specific years in which sampling takes place. If sites are surveyed twice per year detection probabilities above 95% and 99% can be achieved sooner, at three and 5 years respectively, and the associated uncertainty in when thresholds are reached is reduced (Fig. 2). Three surveys per year achieved a detection probability above 95% after only 3 years of monitoring and 99% after 4 years (Fig. 2). If four surveys are conducted each year, 95% and 99% thresholds are reached after 2 and 3 years respectively, and uncertainty in detection is completely ameliorated following 4 years of monitoring (Fig. 2).

## DISCUSSION

Environmental stochasticity strongly impacts long-term monitoring efforts for Reticulated Flatwoods Salamanders. Even after a decade of sampling, the occupancy status of Reticulated Flatwoods Salamanders cannot be confidently determined if only one survey is conducted each year (Fig. 1). Increasing the number of surveys within a year greatly
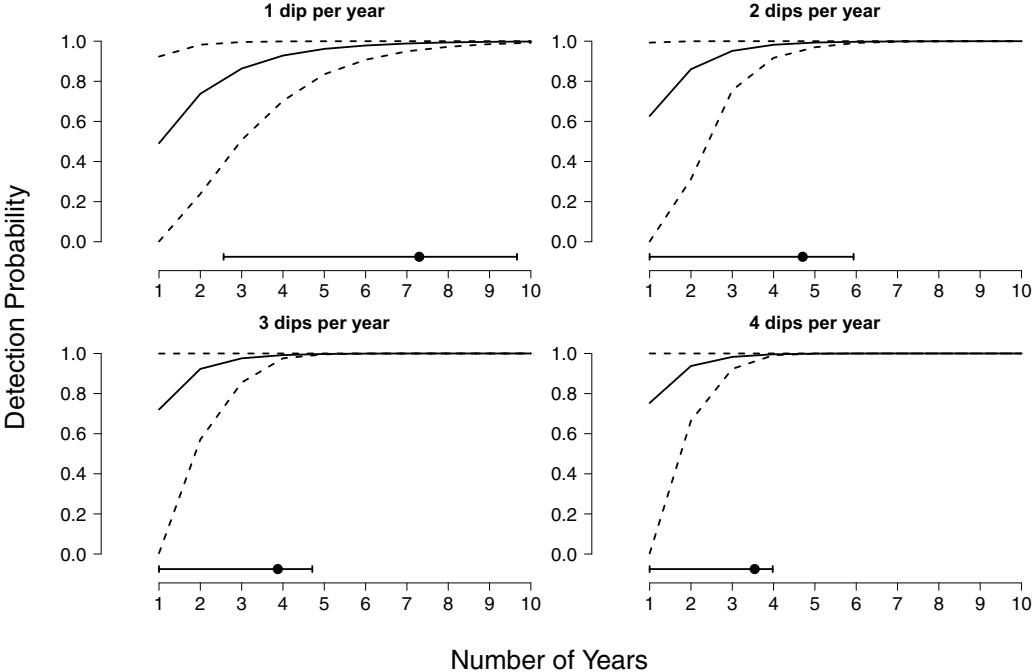

**Figure 2 Cumulative detection probabilities of larval flatwoods salamanders as a function of time and survey effort.** Cumulative detection probabilities are presented for up to 10 years of continuous monitoring. The four panels represent different levels of survey effort within each year (1, 2, 3, or 4 surveys per year). Surveys comprise 30 min of dipnetting effort, typically shared between multiple surveyors, concentrated in areas of herbaceous vegetation. The solid lines indicate median detection probabilities and the dashed lines reflect 90% quantiles (0.05–0.95) given annual fluctuations in detectability. The dot and whisker lines along the bottom of each panel indicate the mean number of years required to achieve a 99% detection probability and the associated uncertainty.

reduces the uncertainty in cumulative detection, but even under the highest sampling intensity, at least 5 years are required before sites can confidently be declared unoccupied (Fig. 2).

Amphibian populations are frequently monitored through dipnet surveys that target larvae following breeding. For Reticulated Flatwoods Salamanders, researchers have surveyed using dipnets two or three times per year (*Palis, 1997*; *Bishop et al., 2006*), and we followed protocols described by *Gorman, Bishop & Haas (2009)*. Standardized monitoring schemes are attractive because they are easy to employ and produce data in a structure desirable for statistical analysis. Such boilerplate schemes however, fail to account for year-to-year fluctuations in survey conditions which prevents a high degree of certainty in predicting site status for low-intensity monitoring schemes. To improve confidence in our own data, we have added a fourth sampling event per season to our protocol, if no flatwoods salamanders were detected after the first three, and have shifted from attempting to sample 25–50% of ponds each year to attempting to sample all ponds annually. This will give us the best chance of having sufficient data over a 10-year period to have confidence in our estimates of occupancy, given there will likely be a few years in each decade when drought makes sampling impossible and likely one or

two years when financial constraints allow us to only sample 50–75% of ponds. Because reticulated flatwoods salamanders have a maximum adult lifespan of around 10–12 years, to ensure that the local breeding population has been extirpated, it is necessary to confirm lack of breeding under suitable environmental conditions over at least a 10-year period. In fact, on our study site, we have observed periods of 8 or 9 years of sampling with no detections and then detected larvae again, in ponds that are far (1,000 m or more) from other occupied ponds.

The complex life-cycle of Reticulated Flatwoods Salamanders severely inhibits a reliable assessment of population viability from aquatic surveys alone. Zero detection can occur if (1) the ponds are already filled with water when adults arrive, resulting in eggs being laid outside the pond basin and therefore never being inundated, (2) the pond never fills or fills too late for hatching to occur, (3) the pond dries before larvae can be detected, (4) ponds are over run with predators. Only when complex facets of a species' life history and ecology are well understood can a flexible monitoring scheme be employed that incorporates the likelihood of encountering an individual in the timing and/or intensity of surveys. Anecdotally, in our own study we have observed a 9-year gap between detections, so could have considered certain populations to have been extirpated if we were not aware of these issues. We had two ponds where flatwoods salamander larvae were detected in 2010, but no detections were made for the next 9 years despite annual sampling. Severe summer drying followed by abundant winter rainfall in 2019–2020 created ideal conditions for larval development and we detected larvae at these two ponds again in the spring of 2020.

Until recently, employing a dynamic sampling approach would make subsequent analysis and detecting long-term trends more challenging, but modern statistical techniques can readily accommodate such imbalanced data. *Charney, Kubel & Eiseman (2015)* describe a method of adaptive sampling that focuses available resources to sample ambystomatid larvae only in years when they are most likely to be detectable, based on conditions in bellwether ponds. Because regional surveys showed high correlation between reproduction within bellwether ponds and other sites, differences in larval abundance were used to determine whether a year was likely to be appropriate for wide scale sampling (N. Charney, 2020, personal communication). However, if breeding attempts are not highly synchronized across the landscape, if bellwether ponds are unavailable, or if climatic conditions that create "good" breeding years are unknown, the approach can be challenging to implement (*Pacifici, Dorazio & Conroy, 2012*; *Shaffer, Roloff & Campa, 2019*).

Although confidence that a species is truly absent if it's not detected can be improved if survey effort is increased, the likelihood will also strongly depend on the prior probability of presence (*Wintle et al., 2012*). This prior could depend on the time since the species was last observed, perceived site suitability, presence of other species at this site or the focal species at neighboring sites on the landscape. Any reintroduction decisions should take into account habitat connectivity, potential future metapopulation dynamics, and long distance dispersal mechanisms. The marked uncertainty in environmental fluctuations

and their impact on detection rates necessitate intensive survey effort over many years before a site can be considered for translocated animals.

For other aquatic organisms for which detection is challenging, environmental DNA (eDNA) has been a powerful tool. The annual fluctuations that make detection through dipnetting challenging however, will likely influence eDNA results in the same way. If the ponds are dry, no method that depends on the presence of aquatic larvae will be able to detect them. When conditions are suitable, even using modified techniques of taking frequent local samples and processing larger volumes of water, our colleagues found that eDNA was comparable to our dipnetting techniques, with both detecting salamanders in most ponds, but each failing to detect salamanders in about the same number of occupied sites (*Goldberg, Strickler & Fremier, 2018*).

Under the ESA, reintroductions are being considered in areas where wild populations have been extirpated to aid recovery of Reticulated Flatwoods Salamanders. In order to identify potential reintroduction sites, intensive, long-term sampling is necessary to provide a high level of confidence that lack of detections reflected true absence. Once new populations have been established, multiple survey methods should be implemented to evaluate the success of reintroduction efforts (*He & Gaston, 2000*; *Manel, Williams & Ormerod, 2001*; *Royle & Nichols, 2003*; *Royle, Nichols & Kéry, 2005*; *Cosentino, 2014*). Owing to density dependence in the aquatic stage, larval abundance can seldom be used to infer population dynamics (*Shea, Wolf & Mangel, 2006*; *Scherer & Tracey, 2011*). Whilst confirming the return of mature individuals to a breeding site, the often weak relationship between larval presence and population processes prevents a robust assessment of long-term viability from dip-net data alone (*Strayer, 1999*; *Gaston et al., 2000*; *Zhou & Griffiths, 2007*; *Conlisk et al., 2009*; *Korfel et al., 2010*; *Bried & Pellet, 2012*).

This paper does not consider cost as a limitation (*Field, Tyre & Possingham, 2005*; *Moore et al., 2014*). It is important to note however, that the cost of larval sampling is orders of magnitude less than the cost of monitoring adults through drift-fence sampling, because the latter requires installation of infrastructure, nightly monitoring at least every 24 h for an extended period of time (or repeated removal and replacement of traps over weeks or months of sampling), and a large enough crew to monitor all sites at the same time. The strategies of stopping surveys after detection (*Regan et al., 2006*; *Rout, Heinze & McCarthy, 2010*; *Guillera-Arroita, Hauser & McCarthy, 2014*) or determining occupancy through eDNA may reduce monitoring costs, but do not readily translate to conservation applications when other management objectives (such as assessing growth rates, obtaining genetic samples from a population, etc.) necessitate captures of multiple individuals.

## CONCLUSIONS

In conclusion, annual variability in environmental conditions can severely affect the results of standardized long-term amphibian monitoring efforts. Ideally, multiple sampling methods would be employed to discern population trends or distinguish between local extirpation and undetected presence, but this is often cost-prohibitive. Given the longevity

of ambystomatid salamanders, coupled with their cryptic lifestyles, a high survey effort across multiple years is required to infer absence, particularly when sites are being evaluated for future reintroductions. The methodologies outlined here will aid other researchers in tailoring monitoring schemes to overcome the confounding effects of environmental stochasticity.

## ACKNOWLEDGEMENTS

We would like to thank the team of people who have made this research possible. Special mention should be given to John Palis, David Bishop, and Thomas Gorman for establishing the dipnetting protocol and database, as well as to Lourdes Oztolaza, Kelly Jones, Brandon Rincon, Jay Parker, Steve Goodman, Vivian Porter, and the myriad technicians involved in data collection. We would also like to thank two anonymous reviewers whose comments greatly improved the manuscript.

### Funding

Over the years financial and logistical support for the project has been provided by Jackson Guard of Eglin Air Force Base, the US Fish and Wildlife Service, Virginia Tech Department of Fish and Wildlife Resources, the Florida Freshwater Fish and Wildlife Conservation Commission and the USDA National Institute of Food and Agriculture, McIntire Stennis project 136627. The funders had no role in study design, data collection and analysis, decision to publish, or preparation of the manuscript.

### Grant Disclosures

The following grant information was disclosed by the authors:
Jackson Guard of Eglin Air Force Base.
US Fish and Wildlife Service.
Virginia Tech Department of Fish and Wildlife Resources.
Florida Freshwater Fish and Wildlife Conservation Commission.
USDA National Institute of Food and Agriculture, McIntire Stennis Project: 136627.

### Competing Interests

The authors declare that they have no competing interests.

### Author Contributions

- George C. Brooks conceived and designed the experiments, performed the experiments, analyzed the data, prepared figures and/or tables, authored or reviewed drafts of the paper, and approved the final draft.
- Carola A. Haas conceived and designed the experiments, performed the experiments, authored or reviewed drafts of the paper, and approved the final draft.

## Data Availability

The raw posterior distributions for year-specific detection probabilities and the code used to perform the analysis and generate the figures are available in the Supplemental Files.

## Supplemental Information

Supplemental information for this article can be found online at http://dx.doi.org/10.7717/peerj.12388#supplemental-information.

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
