# Peer review of "Using historical dip net data to infer absence of flatwoods salamanders in stochastic environments"

_PeerJ, doi:10.7717/peerj.12388_

## Round 0.1 · original submission · Minor Revisions

Thank you for this submission which was well received by both reviewers and myself. Overall, I agree with the reviewers that this is a solid contribution and requires only some minor revisions to be accepted for PeerJ.

In addition to the reviewers' comments, I have the following remarks:
- For me, your stated aim in the introduction is unclear "...a procedure for developing..." (L88), and it doesn't allow you to tightly align your results and discussion (contra Rev#2 add. comm.). I think that this is just a question of reflecting on the original aims (maybe a previous ms or proposal?).

- as noted by the reviewers, life-history information on this species is scant and scattered within the ms. My preference is to condense this into a section of the M&M. Given that this species is so well studied, a short section that gives pertinent life-history information and citations to other relevant studies would be sufficient. (I don't think that this needs to be in the intro).

- please could you make it clear whether this study collected the data, or whether it took the data from another study. The ambiguity was picked up by Rev#2 and I feel that it is a distraction for the reader. Once you've decided, please amend the M&M appropriately.

- today many researchers would approach this same question with eDNA, and I feel that not mentioning this as another approach (e.g. in the discussion) makes it into 'an elephant' in your manuscript. You don't need to justify your approach, but you could state how your approach could be combined with eDNA (which also has drawbacks).

- although you might consider their request, there is no real need to shorten your discussion (as requested by Rev#2)

·

Basic reporting

Overall, this is a very well written article, and the basic reporting is good. The introduction is well structured, but does not introduce the focal species. The discussion is too long for my taste. I suggest moving some of the life history information and previous work on the focal species into the introduction. I have some questions about the methods, including the supplementary script, that I elaborate on in the experimental design section of this review. I found the results clearly presented and easy to understand, with the exception of use of credible intervals, confidence limits, and quartiles. All three are used to summarize findings, and I found this a little confusing. The figures support the findings, and are well designed. I would like to see the authors reiterate a clear recommended survey scheme for their focal species in the conclusions section, if possible.

I have a few very minor comments for the Abstract and Introduction.
Line 30: detection probability is referring to aquatic juveniles and subadults, I think.

Line 34: Is a suitable site for reintroduction any site that is truly absent? If so, I’m not sure this is a major takeaway of your study.

Line 58: Comma after detectability

Line 75: Comma following i.e., throughout the manuscript

Line 88: This paragraph needs to be limited to just thesis statements and hypotheses. Introduction of the focal species should come before here.

Line 89: It seems 8 year’s of detection probabilities are given from the previous study, and subsampled into 10 “years” worth of data. I think maybe some distinction needs to be made between these two things. Maybe the term “pseudo-years” if that isn’t too obnoxious?

Experimental design

This article presents original primary research within the scope of PeerJ. The research questions are relevant to conservation of species that have multiple life stages, with heterogeneous detection probabilities. The investigation was performed to a high ethical and technical standard. The methods were described in enough detail to replicate them, but could benefit from some additional clarification.

Line 112: regarding research approval declarations. Is this declared in the original Brooks et al 2019 article? If so, is it necessary to repeat here, given this study refers only to previously published data? Further, if an IACUC statement is needed, are federal and state permit numbers also missing here? Personally, I think you could get rid of this statement, and refer readers to the previously published work.

Lines 129-131: Does line 52 of the R script demonstrate this? Why is only the first column of each sampled year used here? What are the other 19 for?
In the creation of Pstar1 etc, columns 2:19 seem to refer to repeat sampling? If this step is done here, on randomly drawn samples, why perform this calculation again at lines 56-58 for example? I may be misunderstanding the approach used, but this seems to be where the (1-p)^n formula is first implemented.

Line 134: The investigation of repeat surveys within a single year uses 1000 draws. Your script indicates 2700 draws were conducted for the multi-year simulation.

Lines 134-138: Is this right? or is this meant to be Occupancy probabilities? The wording of this sentence has me totally confused. I think the step referred to in this sentence deserves more explanation. Further, Is there a precedent for this approach available that could be cited here to give readers a deeper understanding of why this is true?

Line 142: R script indicates a 90% credible interval here, and I think that should be stated explicitly.

Regarding the supplementary script. I would like to see some notes spelled out more clearly. There’s a lot of loops used to populate subsamples, I’d like to see some more information given about each step in the process as comments in the script. For example, at lines 49-55 of the script, slightly more information could help a reader move through this supplement faster, making it more useful to novice coders (which I might qualify as!). Here’s what I mean:

I_ran = matrix(nrow = 10, ncol = nreps) #creates 10x1000 matrix
for (i in 1:10) { #populates rows
for(j in 1:nreps) { #populates columns
#each sample is the mean of the first column of a random year
I_ran[i,j] = colMeans(ps[[sample(1:8, 1)]])[1]
}
}

The Rdata file “supp_1” loads an object titled “dat”, but the supplied script utilizes an object titled “outAB”. This wasn’t immediately obvious to me, and it would probably benefit readers to modify one or the other of these supplemental files so they match.

At line 67 the function rowMedians is used. This requires the package matrixStats. I think it’s helpful if readers are given code that initializes all necessary packages to execute example code, at least something like library(matrixStats) or install.packages(“matrixStats”) to tip them off to this type of thing.

I may be missing something, but as near as I can tell, there are sections of code at lines 18-24 and 92-95 that can be removed. These “x” objects don’t seem to be used anywhere else in the script, and they are overwritten by the use of “x” in the generation of your figures starting at line 158.

At line 101, the median of a single number is taken. I think it’s worth removing this for clarity towards the readers. If this method is applied to a group of values elsewhere in the script, then I think it’s alright to keep, but I didn’t notice it being used elsewhere.

Validity of the findings

Without question, the data utilized by the authors is robust and statistically sound. This is a simulation-based study, stemming from previously published observational data, and thus controls are not of concern. The authors have made their data available and for the most part it is reproducible quite easily (although some improvements to the supplementary script are proposed in the previous review section). The conclusions are generally supported by their findings, but may be overstated some. The authors repeatedly refer to identifying suitable reintroduction sites for their focal species, however, their data only enable identifying a method for calculating how much effort is required to determine the species is absent from a site. If absence from a site is all it takes to be suitable for reintroduction, then that’s worth stating in the discussion, but I do not see this as the primary finding of this study.

Regarding the Results section, I have some specific comments.

Line 150: “high detection probability.” This is a vague term (which I have been criticized for using often, myself!). I suggest rewording this to be explicit about what needs to be achieved, whether that’s mean detection probability of above 0.99 or otherwise. If it’s not clear what the goal detection probability is, then this idea needs further explanation.

Line 154: Here you call it a quartile (and I think that makes sense), but in the figure caption it’s called a 75% credible interval, and I don’t think those are the same thing. But, if they are the same thing, use the same name when referring to them throughout the manuscript. Additionally, the R script seems to be calculating a 50% interval in any case.

Lines 155-156: 95% confidence limit. I do not see example of this being calculated in your script. I also think that if you’re going to refer to the 95% confidence limit here, that should also be demonstrated in the figure. Which is more useful for management of the species, or determining the necessary survey effort, quartiles or the 95% confidence limit (or credible interval)? These terms are not interchangeable to my knowledge, and I think if you streamlined your use of them it would spare readers some confusion.

I mentioned in a previous section that I believe the discussion is too long, and could benefit from moving introductory information about the focal species to the introduction. That is my only concern regarding the discussion section.

I have one minor comment at line 245-246: The information here regarding fire-ants sounds like an anecdote from a related study, and if that is the case I would like to see a citation provided.

At line 236 “Conlisk et al, 2008”, is given as 2009 in your reference list.

These articles are not cited in text but appear in your references. Alford 1999, Berven 1990, Chesson and Warner 1981, Dodd 1993, Endangered Species Act, Green 2003, Lande and Barrowclough 1987, Lande 1993, Mehta et al 2007, Nielsen et al 2005, Pechmann et al 1991, Pechman and Wilbur 1994, Rhodes et al 2006, Rhodes and Jonzen 2011, Salvidio 2009, Semlitsch 1983, Semlitsch 1987, Semlitsch et al 1996, Semlitsch et al 2014, Vucetich and Waite 1998, Whiteman and Wissinger 2005.

Finally, the caption of Figure 1 indicates a 75% credible interval, but the R script demonstrates 50% (0.25 and 0.75 quantiles). Likewise, Figure 2 indicates 95%, but quantiles used in the R script are 90% (0.05, 0.95).

Additional comments

I really enjoyed reading this article. It is well written, timely, and focuses on a topic of critical importance to conservation of endangered species. I apologize for agonizing over your provided script, but I found this to be a hugely useful tool for understanding your work. I believe simulation studies of this nature are invaluable towards identifying the best available methods for achieving success in determining the presence and abundance of endangered species within their given landscapes. Congratulations on a wonderful study.

Reviewer 2 ·

Basic reporting

In general, I found this to be a fairly well written manuscript that made use of sufficient background information and citations, used clear and professional English, and adhered to a proper article structure.
I caught a few minor issues with the tables and the textual citations of the second figure.
o Please elaborate on why 75% CIs are used in Figure 1.
o Should Line 180 refer to Figure 2 instead of Figure 3? If not, I was not provided with any Figure 3.
o The caption for Figure 2 requires more detail. Please explain what the dot and whisker lines along the bottom of each panel represent.

I appreciate that the data were shared, and am especially impressed that the code was shared as well. I did, however, find that some necessary details were omitted from the supplemental code document. Namely, the data file was not loaded with the name that was used later in the coding, nor were packages required to run certain functions named. I believe that this would require a simple fix. Adding a setup section to the provided code to a) name the data file as it is read into the work environment and b) load the required packages would greatly improve the access to these supplemental materials. Additional comments would also be useful to align with the methods described in the text.

For the most part, I found this paper fairly clearly written. I would like to call your attention to a handful of places that I found inaccurate or difficult to interpret:
o Line 28-30: Initially I did not follow the logic of this statement. I believe it would be helpful to explain somewhere in the abstract that this is a situation that arises because monitoring of the breeding population occurs indirectly through surveys of a different life stage of the organism.
o Line 218: Should this refer to “time since the species was last observed?” If not, I need more clarification because “the time the species was last observed” is unclear to me in this context.
o Line 242: Does obtaining genetic samples require multiple captures? I can see how growth rates (or possibly checks for disease/parasites) can require multiple captures, but to my knowledge genetic samples are generally collected from an individual organism just once as genetic material should not change over time.

Experimental design

This work is well aligned with the aims and scope of the journal. The stated research question is to present a procedure for developing an appropriate monitoring scheme for a species of salamander that incorporates environmental stochasticity so that confidence in non-detections is sufficiently high that conservationists can select unoccupied sites for reintroductions. The authors also have the goal of developing a procedure that can be adapted for any species with sufficient detection data over a series of years that include a representative sample of environmental conditions.

The information gap that the researchers are seeking to fill is the need for approaches to confidently predict the site status and determine outcomes of reintroduction, given the variability of environmental conditions. This work clearly demonstrates the high degree of uncertainty in detection probability that can result from environmental stochasticity for a species of conservation concern with a highly complex life cycle. This study demonstrates that repeated surveys are required over a number of years for the focal species, and demonstrates how long-term monitoring data can be used to identify the number of surveys per year and years of monitoring required to reach an acceptable level of confidence in site status.

The authors use a robust, hierarchical analysis procedure. I appreciate that the code and data have been shared. The ethical standards were approved by the home institution of the researchers.

While I followed what they did in the code, I would appreciate some additional explanation for Lines 129-131. I could not replicate this from the text alone (say if the link to the supplemental material was broken).

Validity of the findings

The underlying data have been provided and come from many years of monitoring data.
o Please clarify in the text whether you used detection data from 2018 and 2019. You report in the methods that the study lasted from 2009-2019, but the Brooks et al. paper reports year-specific detection probabilities only for the period from 2009-2017. I believe that this 8 year sample provides a sufficiently robust dataset for your bootstrapped analysis, but it needs to be explicitly stated how many years of detection probability data are used.
o I have some concerns about the range of wetland sizes (0.1-20.9 ha). You used a similar amount of sampling effort (30 minutes per visit) at all wetlands, but I would be far more confident calling a non-detection from a 0.1 ha wetland a true absence than a non-detection at a 21 ha wetland.

The conclusions are appropriate and well founded based on the results. However, I found that the discussion of the results was insufficient to match the aims laid out in the abstract and introduction. I would suggest that you spend more time in the discussion section explaining how other researchers can apply this approach to their own species of interest. What are the necessary steps (could you list or label them more clearly)? Is it important to both determine how many years are necessary with one survey per year AND how many years it will take them if effort is increased? How should they decide on a threshold of detection probability that is appropriate for their species? And finally, how will this help you moving forward with monitoring for Reticulated Flatwood Salamanders, and with reintroductions of this species? You state that intensive, long-term sampling is necessary (Line 227), but does that mean that you will recommend 4 surveys a year for five years? Or stick to 2-3 surveys for a longer time period. How can you (and others) weigh the trade-offs of more visits vs. more years? How should researchers respond if they hit a series of bad years mid-project? Addressing some of these questions would greatly increase the utility and usability of your manuscript.

Additional comments

Dear authors, it was a pleasure to read your manuscript. You clearly stated your research question, and wrote persuasively about the need for better monitoring schemes to account for environmental stochasticity, especially in the context of conservation reintroductions. It is an interesting case study of a species with an environmentally-mediated life cycle, and indirect monitoring procedure for the adult population, and a long-term monitoring dataset. You made good use of bootstraping and a hierarchical approach to simulate the amount of monitoring required to achieve confidence in site status, which is an ongoing issue in conservation and monitoring. I would love to see more in the discussion section regarding how this approach can and should be implemented by others, since you state that that is one of your main goals.

---

## Round 0.2 · accepted · Accept

Thanks for your revision. I think that this manuscript is now acceptable for publication in PeerJ.

A couple of points:

1. In your rebuttal you state that you made it "clearer that we were drawing from only 8 years of real data." However, both abstract and later under the statistical approach, you mention 10 years. Ten years also pops up as the time period in the discussion. The fact that it is 8 years of data is only obliquely referenced [subtraction of 2009 from 2017]. Could you make sure that the ns is consistent and clear on this aspect of the data?

2, It would be nice to acknowledge the input of the reviewers.